# Understanding the Spread of Fake News: An Approach from the Perspective of Young People

Alejandro Valencia-Arias [1,2,*] , Diana María Arango-Botero [2], Sebastián Cardona-Acevedo [2,3], Sharon Soledad Paredes Delgado [4] and Ada Gallegos [5]

1 Escuela de Ingeniería Industrial, Universidad Señor de Sipán, Chiclayo 14001, Peru
2 Departamento de Ciencias Administrativas, Instituto Tecnológico Metropolitano, Medellín 050036, Colombia; dianaarangob@itm.edu.co (D.M.A.-B.); sebastianc0617@gmail.com (S.C.-A.)
3 Centro de Investigaciones—CIES, Institución Universitaria Escolme, Medellín 050040, Colombia
4 Facultad de Ciencias Empresariales, Universidad Señor de Sipán, Chiclayo 14001, Peru; paredesd@crece.uss.edu.pe
5 Instituto de Investigación y Estudios de la Mujer, Universidad Ricardo Palma, Lima 15039, Peru; ada.gallegos@urp.edu.pe
* Correspondence: valenciajho@crece.uss.edu.pe; Tel.: +57-3002567977

**Abstract:** The COVID-19 pandemic and the boom of fake news cluttering the internet have revealed the power of social media today. However, young people are not yet aware of their role in the digital age, even though they are the main users of social media. As a result, the belief that older adults are responsible for information is being re-evaluated. In light of this, the present study was aimed at identifying the factors associated with the spread of fake news among young people in Medellín (Colombia). A total of 404 self-administered questionnaires were processed in a sample of people between the ages of 18 and 34 and analyzed using statistical techniques, such as exploratory factor analysis and structural equation modeling. The results suggest that the instantaneous sharing of fake news is linked to people's desire to raise awareness among their inner circle, particularly when the messages shared are consistent with their perceptions and beliefs, or to the lack of time to properly verify their accuracy. Finally, passive corrective actions were found to have a less significant impact in the Colombian context than in the context of the original model, which may be explained by cultural factors.

**Keywords:** new media literacies; new generations; young people; digital behavior; fake news; factor analysis





## 1. Introduction

In the current knowledge-based society, there is a very high probability of encountering fake news, due to the massive flows of information circulating and, sometimes, produced by all types of users [1]. The rapid spread through different communication channels, especially social media, has become a matter of great concern. This situation has been exacerbated by the COVID-19 pandemic, with numerous fake news stories about the origin and transmission of the virus circulating online [2]. People's heavy reliance on social media and digital news portals, where information sources are not always verified, has increased the risk of being misinformed [3].

The effects of spreading fake news can transcend, among others, the social, cultural, political, and economic spheres. In the political context, for example, false information can influence important decisions such as elections, which is very harmful to society [4]. The topics that are most susceptible to misinformation are politics, COVID-19, science and technology, climate change, the competitiveness of globalized economies, vaccinations, and higher education [5]. These topics condition some aspects of life, including employment, income, social inclusion, violence, and the economy.

Authors such as Tandoc, Thomas and Bishop [6] have emphasized the importance of examining social media users' motivations to share fake news. Moreover, ref. [7] studies in this field have primarily focused on the United States and other developed countries, which is why this research was conducted in a Latin American context, specifically Medellín (Colombia). Its purpose is, thus, to contribute empirical evidence on the matter considering a different social, economic, cultural, and political scene.

For young people, social media and the internet have become the primary sources of information when it comes to keeping up with the latest news and trending topics [4]. This, combined with a large amount of false content circulating online and the lack of training to spot it [8], motivated this study, which aims to identify the factors that influence the spread of fake news among young people in Medellín (Colombia). Another relevant reason for this study is that young populations believe that misleading information is most commonly shared among older age groups [5]. This belief, indeed, reveals that young people are unaware of their responsibility for the information found on the internet.

### 1.1. The Spread of Fake News among Young People

Traditional news sources often follow strict codes to verify stories before sharing them, but today, internet users can upload news on social media and unverified portals without proving their accuracy [3]. Hence, social media platforms (whether considered trustworthy or not by their users) have become the main scenario for disseminating content that permeates social realities [1] understanding the lack of veracity of the content that is uploaded on social networks [3].

The spread of fake news is most common among young people, given that they are natural adopters of technology and the internet, which means that they receive and share large amounts of data and news via digital channels [4]. Similarly, other authors suggest that there is currently a growing demand for information, not only because of the pandemic and the uncertainty it has caused, but also because, in some cases, conventional resources are not enough, especially for young people [2]. As a result, young populations have turned to the available media and have become the biggest consumers of social media content [1].

### 1.2. Proposed Model on the Factors That Favor the Spread of Fake News among Young People

Several studies on fake news, its definition, detection, and dissemination have been conducted [9,10]. For example, some authors, have examined the importance of adopting procedures to spot fake news, particularly among populations under twenty years of age [4]. Castellini, Savarese and Graffigna [11] found that algorithms are not enough to stop the spread of misinformation online, as they only help to solve the problem partially. According to these authors, this situation needs to be addressed from a psychological perspective.

Other authors have analyzed some specific factors that explain the phenomenon of fake news sharing: the instantaneous sharing of news for creating awareness, active corrective actions on fake news, passive corrective actions on fake news, sharing fake news due to personal beliefs, and sharing fake news due to lack of time [12]. These authors, for instance, found a positive association between the sharing of fake news and young people's urge to share news instantly due to a psychological need to keep their contacts informed and stay connected.

The lack of time has also been proven to be a factor influencing people's decision to spread false information. Although news can be manually verified using supervised machine learning techniques that integrate metadata with text, this is time-consuming and relies heavily on individuals' subjective decision-making [13]. Concerning the influence of personal beliefs, Marwick [14] explains that, from more political aspects, there are two major reasons why users share fake news: the first is because of the partisan media deception, and the second is because the fake news is aligned with each user's worldview, beliefs, and social positions.

Social media is often used for disinformation purposes that elicit emotional responses from users, whose attention is then monetized [15]. This situation is not alien to the Colom-

bian context and affects the behavior and decision-making power of young generations. Among this population, it is worth mentioning millennials, who represent the majority of the workforce in the world and whose decisions, thus, have a great impact on all spheres of society, especially since they are considered digital natives and heavily rely on digital media to stay informed.

According to various studies, media users usually believe that the information shared by their relatives, friends, and personal references is reliable [16]. Likewise, according to another study [10], there are different predictors of individuals' susceptibility to believe fake news in political contexts, such as message characteristics, belief consistency, and presentation cues.

Importantly, the phenomenon of fake news is a recent field of research (Wang et al. [17]). In fact, it first appeared in the literature in 2005 in a study by Pavlik [18]. Only in 2012, researchers started to show a real interest in this topic, and six papers were published during that year. However, between 2016 and 2021, there was a proliferation of studies on the subject, with about 641 publications by the end of this period. These data suggest that this is a growing field of study with enormous potential to be explored in various contexts.

In Medellín (Colombia), some studies have been identified, such as those by Cortés Vela [19], which discuss scientific activity in times of fake news in the context of the city; however, the authors do not identify studies that address the issue from the perspective of the dissemination of fake news among young people. Therefore, this paper seeks to contribute knowledge on the spread of fake news in Latin America, particularly among the young population of Medellín, and the factors that influence it. For this purpose, we applied the model proposed by Talwar et al. [12], considering that it has been attracting the interest of the scientific community and that the relationships between constructs and the items to measure them have been empirically validated. Such a model was constructed using data from a metropolitan city in India, presentando el instrumento por medio del cual se recopiló la información. This allowed us to replicate the model, just as Wang, Zhao and Chen [20] did, to create a system for detecting fake news related to COVID-19.

As mentioned above, research on the spread of fake news is still in its early stages, particularly in Latin America and other developing countries, where social media is more widely used [21]. Also, 83% and 82% of the people with access to the internet in Central and South America, respectively, use social media [22]. These are the two most representative percentages worldwide; hence the relevance of conducting empirical studies on this phenomenon.

Such studies may help young users to be aware of the factors associated with the spread of fake news and thus prevent them from believing them and help reduce their spread. In addition, they will be key for organizations, brands, service providers, professionals, and researchers to detect false information, be careful about their contents, and redefine the way news stories are shared to avoid ambiguities. Finally, they could also add knowledge on other unexplored aspects of social media user behavior, which has repercussions on the social, economic, political, and other spheres.

## 2. Materials and Methods

### 2.1. Research Model

The model to be fit is described in Figure 1.

Structural Equation Modeling (SEM) was used to fit the model and test the proposed hypotheses. Importantly, SEMs are composed of a measurement model (relationships between constructs and their associated items) and a structural model (inter-construct relationships). Before performing SEM, it was determined whether the application of the factor analysis was appropriate using the Kaiser–Meyer–Olkin (KMO) test and Bartlett's test of sphericity.

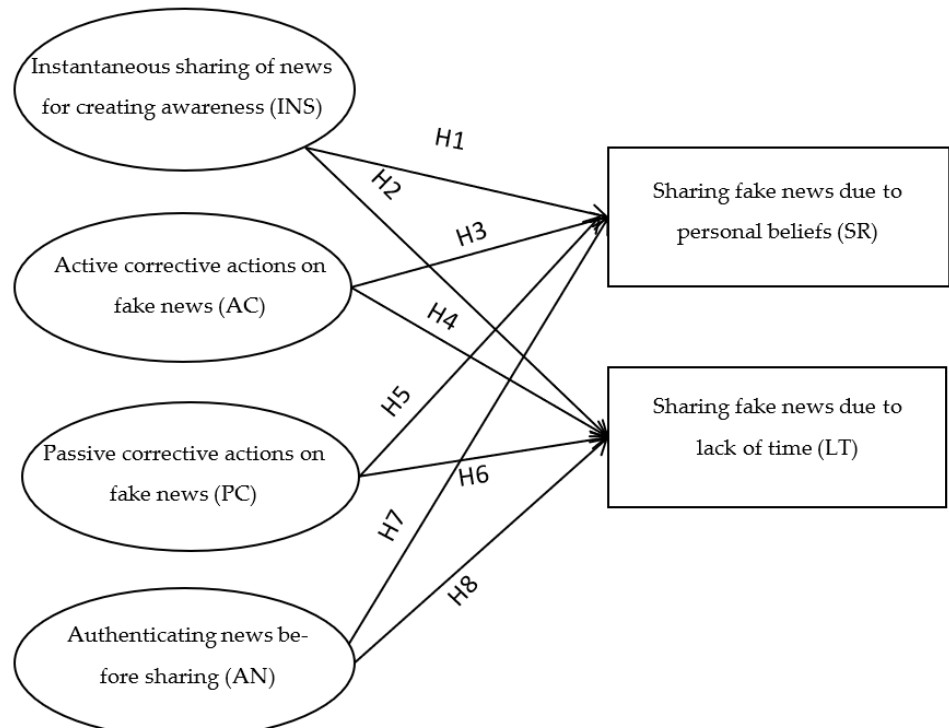

**Figure 1.** The research hypotheses of the proposed model.

*2.2. Research Hypotheses*

**Hypotheses 1 (H1).** INS has a positive association with SR.

**Hypotheses 2 (H2).** INS has a positive association with LT.

**Hypotheses 3 (H3).** AC has a negative association with SR.

**Hypotheses 4 (H4).** AC has a negative association with LT.

**Hypotheses 5 (H5).** PC has a negative association with SR.

**Hypotheses 6 (H6).** PC has a negative association with LT.

**Hypotheses 7 (H7).** AN has a negative association with SR.

**Hypotheses 8 (H8).** AN has a negative association with LT.

*2.3. Variables*

The research model consists of four independent variables: instantaneous sharing of news for creating awareness (INS), active corrective actions on fake news (AC), passive corrective actions on fake news (PC), and authenticating news before sharing online (AN). Since they are constructs or variables that are not directly observable, we used some of the items proposed by Talwar et al. [12] to measure them: four items to measure INS, five items to measure AC, four items to measure PC, and five items to measure AN. Each item had five response options ranging from 1 (strongly disagree) to 5 (strongly agree).

Additionally, the model includes two endogenous variables: sharing fake news due to lack of time and sharing fake news due to personal beliefs. To measure these variables, participants were asked to indicate (on a scale with response options) their level of agreement with the following two statements: (i) I share fake news because I do not have time to check the accuracy of the information, and (ii) I share news that is aligned with my beliefs, even though they may turn out to be fake.

### 2.4. Sample

A questionnaire was sent via email to young people in Medellín between October and November 2021. Convenience sampling was used because it does not limit the sample size [23] and, therefore, a greater amount of data can be collected, which is ideal for exploratory studies [24]. A total of 414 fully completed questionnaires were received, 10 of which were excluded from the analysis due to their constant, monotonically increasing, or monotonically decreasing behavior.

The sample characterization revealed that 63% of the respondents were women. In terms of age, 50% were between 18 and 25 years old; 33% were between 26 and 33 years old; and the rest were over 34 years old, with the understanding that a young adult is considered up to the age of 35 [25]. As for the level of education, 29% held a three-year associate degree; 22% had a high-school degree; 21% had an undergraduate degree; 20% had a two-year associate degree; and the rest had a graduate degree.

## 3. Results

Following the age distribution of the sample and intending to find a summary measure for each construct from a series of items, we proceeded with the test of equality of medians between the different age groups. The Kruskal-Wallis test was used because of the ordinal nature of the dependent variables and because there are more than two groups to compare. The $p$-values for each of the hypotheses suggest non-rejection of equality of medians at a 5% significance level ($p$-value = 0.2254 for the AC construct, $p$-value = 0.07143 for INS, $p$-value = 0.5702 for LT, $p$-value = 0.5063 for PC, $p$-value = 0.6089 for AN and $p$-value = 0.8971 for SR). The above suggests that there are no significant statistical differences in the way of responding according to age groups and, therefore, it is feasible to unify the responses.

To assess the quality of the instrument and the items' adequacy to measure each of the proposed constructs (independent variables), we performed the KMO test and Bartlett's test of sphericity. The KMO value obtained was 0.892, and the significance level of Bartlett's test was 0.000, which suggests that the data were suitable for factor analysis.

An Exploratory Factor Analysis (EFA) was first carried out to examine the underlying structure of the items. Table 1 presents the rotated component matrix, which includes four components that explain a total variance of 57.823%. According to the results, items AN1, AN2, AN3, AN4, and AN5 are problematic because they show correlations above 0.4 with different components, which may later translate into a lack of convergent and discriminant validity. For this reason, the AN variable was excluded from the analysis, as well as the hypotheses related to this construct.

**Table 1.** The rotated component matrix—EFA results.

| | Component | | | |
|---|---|---|---|---|
| | 1 | 2 | 3 | 4 |
| INS1 | 0.355 | | 0.63 | |
| INS2 | | | 0.782 | |
| INS3 | | | 0.762 | |
| PC1 | | 0.759 | | |
| PC3 | | 0.686 | | |
| PC4 | | 0.696 | | |
| INS4 | 0.568 | | | |
| AC1 | 0.748 | | | |
| AC2 | 0.74 | | | |
| AC3 | 0.651 | | | |
| AC4 | 0.722 | | | |
| AC5 | 0.736 | | | |
| AN1 | 0.493 | | | 0.589 |

**Table 1.** *Cont.*

| | Component | | | | |
|---|---|---|---|---|---|
| AN2 | 0.645 | | | | |
| AN3 | 0.64 | | | | 0.477 |
| AN4 | | | 0.312 | 0.337 | 0.594 |
| AN5 | | | | 0.439 | −0.643 |
| PC2 | 0.306 | | 0.577 | | |

Note: The extraction method used in this study was a principal component analysis, and the rotation method we used was varimax with Kaiser normalization. The rotation converged in seven iterations.

Importantly, in order to test the hypotheses, the measurement model (relationships between constructs and their associated items) must be previously validated.

Subsequently, we assessed the internal consistency, convergent validity, and discriminant validity of the three constructs suggested by the EFA. Table 2 shows the results obtained after eliminating three problematic items (INS4, PC3, and PC4). As observed, all the constructs obtained a CR value above 0.7 (internal consistency), an AVE value above 0.5 (convergent validity), and an MSV value below the AVE value (discriminant validity) [26], which confirms construct validity (relationships between constructs and their associated items). Moreover, as can be seen in the last columns of the table, the correlation coefficients (values outside the diagonal) are lower than the square root of the AVE (values on the diagonal), which confirms discriminant validity.

**Table 2.** The validity and reliability measurements.

| | CR | AVE | MSV | PC | INS | AC |
|---|---|---|---|---|---|---|
| PC | 0.706 | 0.546 | 0.433 | 0.739 | | |
| INS | 0.754 | 0.507 | 0.176 | 0.420 | 0.712 | |
| AC | 0.844 | 0.519 | 0.433 | 0.658 | 0.368 | 0.721 |

Once the measurement model was validated, the structural component (research hypotheses) was verified using SEM. The SEM results showed a good model fit (Chi-square/df = 2.117, CFI = 0.97, TLI = 0.958, RMSEA = 0.053, SRMR = 0.052) [27]. After evaluating the model fit, we tested the hypotheses, all of which were accepted at a significance level of 5%, except for those related to PC. On the one hand, the relationship between PC and SR was not found to be significant, and, on the other hand, the relationship between PC and LT was found to be significant but positive (positive beta: 0.260) (see Figure 2).

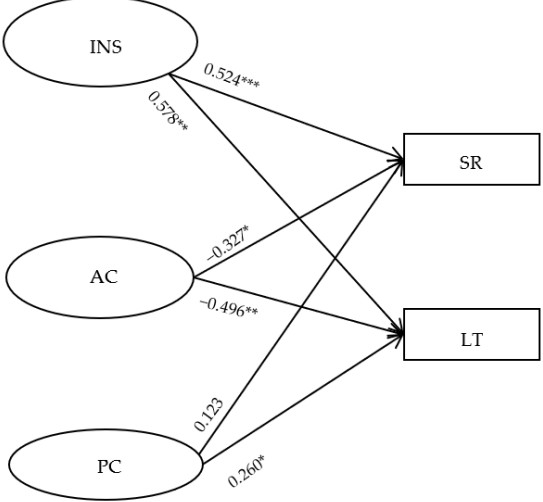

**Figure 2.** A structural model was proposed after validating the measurement model. * $p < 0.05$, ** $p < 0.01$, *** $p < 0.001$.

## 4. Discussion

Fake news detection, which has been studied for some decades now, has become more relevant today as a result of the rumors circulating on social media about the COVID-19 pandemic. This topic can be analyzed using psychometric techniques, such as SEM, which allows latent or unobservable traits to be measured, as well as the relationships between them, thus making great contributions to the fields of cognitive, social, and political psychology [28].

The present study confirmed the proposed correlation between INS and SR, which was found to be the second strongest correlation within the model. This result is similar to the findings of Talwar et al. [12], who reported it as the most important association. According to this finding, in both contexts, users share the news instantly to seek to inform their inner circle, especially when the information happens to be aligned with their personal beliefs. This assertion, in turn, proves that individuals share the news if it is consistent with their beliefs and opinions [14]. This is why authors such as Shoemaker [29] highlight the need for young people to cultivate media literacy to be well-informed about topics related to their personal beliefs.

Additionally, the relationship between INS and LT was found to be the most important in our analysis and the second most important in the original model by Talwar et al. [12], which proves the validity of the proposed hypothesis. This means that, as suggested by Vicente-Domínguez, Beriain-Bañares and Sierra-Sánchez [16], social media users instantly share news that they believe may have an educational impact on their inner circle, without checking their veracity first due to a lack of time. This is also in line with the findings of Pennycook and Rand [28], who claim that people often fail to discern truth from falsehood in the news that they share and do not stop to reflect on their accuracy.

The negative correlation between AC and SR is also supported. In this regard, the empirical evidence shows that active corrective actions on fake news can reduce the amount of misleading information shared due to personal beliefs, provided that these actions involve communicating with those sharing misinformation [30] to inform, advise, recommend, or educate.

Another negative correlation we identified was the one between AC and LT, which helps us understand that the more active corrective actions are taken, the lower the volume of false news shared due to lack of time. This is consistent with the ideas of Cohen et al. [30], who state that active corrective actions are aimed at protecting close people from the influence of fake news, especially within the context of social media. This result was also confirmed in the original paper by Talwar et al. [12], who reported a significant negative correlation between the two variables.

The two hypotheses related to PC were not supported. First, the proposed negative correlation between PC and SR was not found to be significant in our analysis, which suggests that passive corrective actions on fake news have no relevant impact on the sharing of fake news due to personal beliefs. Authors such as Monteiro et al. [31], however, emphasize the importance of these measures when it comes to minimizing the (economic) losses caused by fake news to individuals and organizations. Second, the proposed correlation between PC and LT was negative in the original study by Talwar et al. [12]. Nevertheless, it was found to be positive in our analysis. Hence, future studies should further investigate the association between passive corrective actions (such as reporting or blocking those who share fake news) and the sharing of fake news due to the lack of time to verify their accuracy.

Although various authors have implemented diverse psychometric models to understand the dynamics of fake news and disinformation in society, they have focused on general aspects such as culture, education, age, or gender. For other authors, for example, the most influential factor in the acceptance of fake news is age [32]. For their part, Castellini, Savarese and Graffigna [11] analyzed the spread of fake news in the Italian context, with an emphasis on misinformation about food. Also, authors such as Yang and Tian [2] have applied the theory of planned behavior to understand the effect of recent fake news about COVID-19.

Finally, it should be noted that among the limitations of the research is the fact that the sample is not representative of the population of young people in the city of Medellin. However, we have already mentioned the importance of this research in terms of providing empirical evidence for the validity of some constructs with their respective items and the identification of some factors that have a degree of association with the sharing of false news, which is of interest for people so they can access truthful information that allows them to make effective decisions, regardless of the parameter being analyzed [33], and, on the other hand, that only valuable information becomes viral [34]. This is particularly important considering the impact of disinformation nowadays, which can even permeate political communications often disguised as legitimate, and thus the growing need for fact-checking [35].

## 5. Conclusions

Social networks and the Internet are now widely used to obtain and share information. Among what is commonly shared is news, and according to some studies, there is a high probability it could be fake news. Therefore, different studies, including this one, have been oriented to identify two aspects of the sharing of fake news: 1. What factors are associated with such behavior? 2. What is the relationship between them?

Finding empirical evidence to answer these two questions provides valuable inputs, not only to fill the current theoretical gap but to help young users to be aware of the factors associated with the spread of fake news and thus prevent them from believing them and helping reduce their spread. In addition, they will be key for organizations, brands, service providers, professionals, and researchers to detect false information, be careful about their contents, and redefine the way news stories are shared to avoid ambiguities. Finally, they could also add knowledge on other unexplored aspects of social media user behavior, which has repercussions on the social, economic, political, and other spheres.

Results suggest the instantaneous sharing of news for creating awareness (INS) was found to be one of the main factors influencing the sharing of fake news due to both personal beliefs (SR) and lack of time (LT). This is consistent with the results obtained in the original model and confirms the theory. According to this, the instantaneous sharing of fake news, in both contexts, is associated with, on the one hand, people's desire to inform their inner circle, particularly when the messages are aligned with their values, perceptions, and beliefs, and, on the other hand, with the lack of time to properly verify their accuracy.

Additionally, we found no negative association between passive corrective actions and the sharing of fake news due to personal beliefs. Despite this, and leaving room for further discussion, the model could be applied to different contexts to better understand the role of both passive and active corrective actions in the dissemination of fake news.

Finally, the contribution of this manuscript lies in the identification of variables that have an impact on the spread of fake news among young people, as well as psychological patterns that favor misinformation and transcend different areas of society. Therefore, media professionals should focus their efforts on counteracting the tendency of young users to invest little time, effort, and money in verifying the information's veracity they share, to reduce the risk of sharing false information.

**Author Contributions:** Conceptualization, A.V.-A., D.M.A.-B., S.C.-A., S.S.P.D. and A.G.; methodology, A.V.-A., D.M.A.-B. and S.C.-A.; software, D.M.A.-B.; validation, A.V.-A., S.S.P.D. and A.G.; formal analysis, A.V.-A. and D.M.A.-B.; investigation, A.V.-A., D.M.A.-B. and S.C.-A.; resources, S.C.-A., S.S.P.D. and A.G.; data curation, A.V.-A. and D.M.A.-B.; writing—original draft preparation, A.V.-A., D.M.A.-B. and S.C.-A.; writing—review and editing, S.C.-A., S.S.P.D. and A.G.; visualization, A.V.-A.; supervision, A.V.-A., S.S.P.D. and A.G.; project administration, A.V.-A., S.S.P.D. and A.G.; funding acquisition, A.V.-A., S.S.P.D. and A.G. All authors have read and agreed to the published version of the manuscript.

**Funding:** This research was funded by Institución Universitaria Escolme. The APC was funded by Universidad Señor de Sipán—USS.

**Informed Consent Statement:** Informed consent was obtained from all subjects involved in the study.

**Data Availability Statement:** The data may be provided free of charge to interested readers by requesting the correspondence author's email.

**Conflicts of Interest:** The authors declare no conflict of interest.

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
