# Peer review of "Understanding the Spread of Fake News: An Approach from the Perspective of Young People"

_informatics, doi:10.3390/informatics10020038_

Round 1

Reviewer 1 Report

We begin by congratulating the authors for the proposal, to whom we leave some recommendations:

- Adapt the text so that “According to [3]” does not appear. We are sure it will make reading easier.

- “(…) the topics that are most susceptible to misinformation are COVID-19, science and technology, climate change, competitiveness of globalized economies, vaccination, and higher education. These topics condition some aspects of life, including employment, income, social inclusion, violence, and economy.” (p. 1). Please explain context. We find the absence of the political theme strange, as it is the one that most mobilizes misinformation.

- “For young people, social media and the internet have become the primary sources of 45 information” (p. 2). Need a source for this.

- “As reported by [3], traditional news sources used to follow (…)” (p. 2). Used? Are not currently used?

- “(…) have become the main scenario for 57 disseminating contents that permeate social realities [1].” (p. 2). Please explain.

- “As a result, young populations 64 have turned to the available media and have become the biggest consumers of social media content” (p. 2). Need a source for this.

- “(…) authenting news before sharing online (…)” (p. 2). This doesn’t explain sharing.

- “Concerning the influence of personal beliefs, [14] explains that users share fake news not only because they have been misled by partisan media, but also because those fake news may be aligned with their worldviews, social positions, and beliefs” (p. 3). Please rephrase.

- “According to various studies, young media users usually believe (…)” (p. 3). Only young or all users?

- “In Medellín (Colombia), no studies into this matter have been developed” (p. 3). Can author really assume this? Wouldn't it be more correct to say that the authors did not find any studies on the subject?

- “Social media is often used for disinformation purposes that elicit emotional responses from users, whose attention is then monetized [19]. This situation is not alien to the Colombian context and affects the behavior and decision-making power of young generations. Among this population, it is worth mentioning millennials, who represent the majority of the workforce in the world and whose decisions, thus, have a great impact on all spheres of society, especially since they are considered digital natives and heavily rely on digital media to stay informed” (p. 3). We believe that this paragraph can be repositioned, namely as the third paragraph of this page (3).

- “(…) and the rest, over 34 years old” (p. 5). What authors assume as young people? Until wich age are people young? Please clarify.

- “Authors such as [27] have emphasized the importance of examining social media users’ motivations to share fake news. Moreover, [28] state that studies in this field have 229 primarily focused on the United States and other developed countries, which is why this research was conducted in a Latin American context, specifically Medellín (Colombia). Its purpose is, thus, to contribute empirical evidence on the matter considering a different social, economic, cultural, and political scene” (p. 7). This could be in the first part (introduction).

- “Second, the proposed negative correlation between PC and LT was found to be significant but positive in our analysis” (p. 8). Why positive the analysis? Please explain better.

- “The primary goal, on the one hand, is that people can access truthful information that allows them to make effective decisions, regardless of the parameter being analyzed [33], and, on the other hand, that only valuable information becomes viral [34]” (p. 8). This was not the paper’s  goal, was it?

- “The objective is to build knowledge on the basis of real and proven facts and not on information manipulated by particular interests” (p. 9). In our opinion, can be removed.

Author Response

March 07, 2023

Dear

Informatics Editorial Team

MDPI

Kind regards

According to the suggestions of our article (Understanding the spread of fake news: an approach from the perspective of young people) by the reviewer, the following changes were made, properly marked with red letters in the manuscript:

Reviewer

Comment

Response

R1

Adapt the text so that “According to [3]” does not appear. We are sure it will make reading easier.

The format of the citations has been adjusted to facilitate the reading of the article, according to the reviewer's indications.

R1

“(…) the topics that are most susceptible to misinformation are COVID-19, science and technology, climate change, competitiveness of globalized economies, vaccination, and higher education. These topics condition some aspects of life, including employment, income, social inclusion, violence, and economy.” (p. 1). Please explain context. We find the absence of the political theme strange, as it is the one that most mobilizes misinformation.

The political issue is added as the main source of disinformation, supporting the previous statement on the incidence of false information in terms of decisions and elections.

R1

“For young people, social media and the internet have become the primary sources of 45 information” (p. 2). Need a source for this.

The reference is added to support the statement.

R1

“As reported by [3], traditional news sources used to follow (…)” (p. 2). Used? Are not currently used?

The sentence is modified to account for the processes followed to date by the traditional media, according to the reviewer's questions.

R1

“(…) have become the main scenario for 57 disseminating contents that permeate social realities [1].” (p. 2). Please explain.

An explanatory sentence is added, justifying that this is the main scenario for disseminators of fake news, thanks to the lack of veracity in the content uploaded to social networks, in contrast to traditional media that follow strict verification protocols.

R1

“As a result, young populations 64 have turned to the available media and have become the biggest consumers of social media content” (p. 2). Need a source for this.

The respective citation is added to support the statement.

R1

“(…) authenting news before sharing online (…)” (p. 2). This doesn’t explain sharing.

The sentence is eliminated

R1

“Concerning the influence of personal beliefs, [14] explains that users share fake news not only because they have been misled by partisan media, but also because those fake news may be aligned with their worldviews, social positions, and beliefs” (p. 3). Please rephrase.

The sentence is reworded to make it clearer about what is intended to be said

R1

“According to various studies, young media users usually believe (…)” (p. 3). Only young or all users?

The sentence is adjusted, as the statement applies to all users, not just young people.

R1

“In Medellín (Colombia), no studies into this matter have been developed” (p. 3). Can author really assume this? Wouldn't it be more correct to say that the authors did not find any studies on the subject?

The sentence is adjusted to indicate that although there are some studies on fake news in Medellín, the authors did not identify other studies that address the problem of the dissemination of fake news from the perspective of young people.

R1

“Social media is often used for disinformation purposes that elicit emotional responses from users, whose attention is then monetized [19]. This situation is not alien to the Colombian context and affects the behavior and decision-making power of young generations. Among this population, it is worth mentioning millennials, who represent the majority of the workforce in the world and whose decisions, thus, have a great impact on all spheres of society, especially since they are considered digital natives and heavily rely on digital media to stay informed” (p. 3). We believe that this paragraph can be repositioned, namely as the third paragraph of this page (3).

The paragraph is repositioned, according to the reviewer's indications.

R1

“(…) and the rest, over 34 years old” (p. 5). What authors assume as young people? Until wich age are people young? Please clarify.

Reference is added to justify that persons up to 35 years of age are considered as young people.

R1

“Authors such as [27] have emphasized the importance of examining social media users’ motivations to share fake news. Moreover, [28] state that studies in this field have 229 primarily focused on the United States and other developed countries, which is why this research was conducted in a Latin American context, specifically Medellín (Colombia). Its purpose is, thus, to contribute empirical evidence on the matter considering a different social, economic, cultural, and political scene” (p. 7). This could be in the first part (introduction).

The element is extracted from the discussion and added in the introductory section.

R1

“Second, the proposed negative correlation between PC and LT was found to be significant but positive in our analysis” (p. 8). Why positive the analysis? Please explain better.

The wording of this sentence has been adjusted to mention that in the original study, proposed by Talwar et al., the relationship between PC and LT was negative and that, in this study, the relationship was positive.

R1

“The primary goal, on the one hand, is that people can access truthful information that allows them to make effective decisions, regardless of the parameter being analyzed [33], and, on the other hand, that only valuable information becomes viral [34]” (p. 8). This was not the paper’s  goal, was it?

The wording is adjusted so that the sentence does not confuse with the objective of the research.

R1

“The objective is to build knowledge on the basis of real and proven facts and not on information manipulated by particular interests” (p. 9). In our opinion, can be removed.

The sentence is eliminated

R2

In the summary, the age of the sample should be indicated as 18-34 years.

The respective clarification is added to the summary of the item

R2

It is difficult to understand why India is suddenly alluded to in the summary if it is not explained in this part why it has been taken as a reference or if it is not made clear that a study carried out there is being replicated. As for the findings revealed in this section, it is too obvious to say that there are differences between the results from Colombia and India for cultural reasons. This is usually the norm, especially when comparing two such different realities. In this sense, a very vague claim is also made, already within the study, that India is more religious than Colombia, without supporting this claim with any reference. This does not contribute to the research; a cause-effect relationship with the results is extracted from this idea that is not coherent.

the focus within the manuscript was changed according to the reviewer's comments. Mention is only made of India to point out that the model used as a basis for the study was validated in India. This seems important to us to make clear to the reader the context in which the instrument was developed and to make it clear that it is based on items that have already been validated to measure some constructs or latent variables. For this reason, we also sought a context that was not very different from the Colombian context, which contributes to the validity of the construct.

R2

As for the theoretical framework, it is worth clarifying in line 67 (page 2 of 11), why the term misinformation is chosen here when it refers to the errors inherent in journalistic praxis, there is no intention to deceive young people or anyone else. They are erroneous interpretations, the result of haste or a mistake, as the experts have agreed. It is part of the information disorder, but it does not do the same damage as other variants such as disinformation or malinformation.

This idea and some additional ones were reviewed so as not to generate confusion with concepts such as disinformation and deception. Special attention was paid to ensure that the discourse revolved around the sharing of false news.

R2

It is also generalised when it is said that there are no studies in Colombia on the issue addressed here. They may be incipient, but there are, and they have not been referenced. They refer to young people and the consumption of information or exposure to disinformation, and I think they should be considered (there are linked above).

The sentence is adjusted to indicate that although there are some studies on fake news in Medellín, the authors did not identify other studies that address the problem of the dissemination of fake news from the perspective of young people.

R2

In line 102, you have to say where these figures are taken from. Are they from a systematic review? You cannot list these data without explaining where they come from; it is not very rigorous and not at all scientific.

The source from which the statement is extracted is added, alluding to a systematic review of literature

R2

As far as the sample is concerned, it should be made clear how representative it is of the population of these ages in Medellín.

the sample is not representative and hence it was placed at the end of the discussion section, that the results cannot be generalized and that the contribution of the research is oriented towards the validity of some items and constructs, and towards the identification of some factors that have a degree of association with the sharing of false news, which is of interest to decision makers at governmental levels.

R2

In the presentation of the results, although the hypotheses have been formulated to develop a predictive model, it would be useful, even if the hypotheses have been formulated to develop a predictive model, to break down specific data for each age group in the sample; it is not the same how 18 year olds respond as 34 year olds.

In accordance with the age distribution of the sample and with the aim of finding a summary measure for each construct from a series of items, we proceeded to test the equality of medians between the different age groups. The Kruskal Wallis test was used for this test because of the ordinal nature of the dependent variables and because there are more than two groups to compare. The p-values for each of the hypotheses suggest non-rejection of equality of medians at a significance level of 5%. This suggests that there are no significant statistical differences in the way of responding according to age groups and therefore, it is feasible to unify the responses.

R2

The conclusions lack a reinforcement of the social implications of the model. If this research is inspired by or replicates other research, it should be taken further. What problems is misinformation causing among young people in Colombia and what does this study contribute in this regard?

the evaluator's suggestion was heeded and emphasis was placed on the implications of the results obtained in the research.

We look forward to your comments and hope to hear from you soon.

Thank you very much

_

The authors

Reviewer 2 Report

The research is of potential interest, as indicated at some point in the text, because of its novelty in the South American context. 

In the summary, the age of the sample should be indicated as 18-34 years.

It is difficult to understand why India is suddenly alluded to in the summary if it is not explained in this part why it has been taken as a reference or if it is not made clear that a study carried out there is being replicated. As for the findings revealed in this section, it is too obvious to say that there are differences between the results from Colombia and India for cultural reasons. This is usually the norm, especially when comparing two such different realities. In this sense, a very vague claim is also made, already within the study, that India is more religious than Colombia, without supporting this claim with any reference. This does not contribute to the research; a cause-effect relationship with the results is extracted from this idea that is not coherent.

As for the theoretical framework, it is worth clarifying in line 67 (page 2 of 11), why the term misinformation is chosen here when it refers to the errors inherent in journalistic praxis, there is no intention to deceive young people or anyone else. They are erroneous interpretations, the result of haste or a mistake, as the experts have agreed. It is part of the information disorder, but it does not do the same damage as other variants such as disinformation or malinformation. 

It is also generalised when it is said that there are no studies in Colombia on the issue addressed here. They may be incipient, but there are, and they have not been referenced. They refer to young people and the consumption of information or exposure to disinformation, and I think they should be considered (there are linked above). 

In line 102, you have to say where these figures are taken from. Are they from a systematic review? You cannot list these data without explaining where they come from; it is not very rigorous and not at all scientific. 

As far as the sample is concerned, it should be made clear how representative it is of the population of these ages in Medellín. 

In the presentation of the results, although the hypotheses have been formulated to develop a predictive model, it would be useful, even if the hypotheses have been formulated to develop a predictive model, to break down specific data for each age group in the sample; it is not the same how 18 year olds respond as 34 year olds. 

The conclusions lack a reinforcement of the social implications of the model. If this research is inspired by or replicates other research, it should be taken further. What problems is misinformation causing among young people in Colombia and what does this study contribute in this regard?

Author Response

(The authors gave the same response as above.)

Round 2

Reviewer 1 Report

Thanks to the author for the improvement made. 

Reviewer 2 Report

Thanks for the changes and suggestions done that improve the quality of the research and facilitate it comprehension.